# Analysis on Recycling Channel Selection of Construction and Demolition Waste in China from the Perspective of Supply Chain

**DOI:** 10.3390/ijerph19094975

**Published:** 2022-04-20

**Authors:** Ruwen Tan, Xiang Qing, Jingsong Yang, Jing Zhang, Deng Li

**Affiliations:** College of Architecture and Environment, Sichuan University, Chengdu 610065, China; tanruwen@scu.edu.cn (R.T.); qingxiangscu@163.com (X.Q.); yangjingsong@stu.scu.edu.cn (J.Y.); zhang_jing_scu@scu.edu.cn (J.Z.)

**Keywords:** construction and demolition waste, dual-channel recycling, supply chain, game theory, governmental intervention

## Abstract

Although construction and demolition waste (C&D waste) recycling has been widely regarded as an effective way to save resources, its selection of recycling channels has received little attention. In order to improve the recycling efficiency of C&D waste and promote the process of C&D waste management, we innovatively propose a dual-channel recycling problem of C&D waste from the perspective of supply-chain operation, aiming to study the impact of the selection of recycling channels and the government’s economic intervention on pricing decisions. Specifically, we build in this paper a three-echelon construction material supply chain including a construction contractor, a professional recycling agency, and a building materials manufacturer, considering two modes: the construction contractor does the recycling job alone (as the direct channel) and the construction contractor entrusts the recycling job to the professional recycling agency (as the indirect channel). We use game theory to analyze the optimal decision strategies of the members with or without governmental intervention and the equilibrium strategies are obtained. At the same time, taking Chongqing city as an example, we apply the model to carry out numerical simulation, and the results reveal that greater recycling costs of C&D waste leads to lower profits for the members. When the market capacity of first-class renewable building materials increases, the recycler’s recycling cost of C&D waste decreases, and the remanufacturing cost of recycled materials processed into first-class renewable building materials decreases, the supply-chain members will choose the indirect channel to perform the recycling job. In the case when government intervention exists, the recycling quantity of C&D waste increases, the price of the final products decreases, and both the total profit of the system and the profit of the supply-chain members increase; there is a positive correlation with the government subsidies. The study provides some insights on managerial significance to C&D waste recycling management.

## 1. Introduction

With the rapid development of China’s social economy and the acceleration in urbanization, the amount of construction and demolition waste (C&D waste) generated by construction activities also increasing yearly [1]. At present, most of the C&D waste in China is disposed of by traditional open-air storage or landfill, which not only causes serious damage to the ecological environment, but also threatens people’s physical and mental health [2]. If C&D waste can be effectively recycled, resource utilization efficiency can be improved, the environment can be protected, and the amounts of new resources and manufacturing costs can be reduced at the same time [3], which means recycling and remanufacturing of C&D waste is a win–win situation in the dimensions of environment, economy, and society. “The producer payment system”, “the producer responsibility extension system” in Western countries [4], and the “polluter payment system” and “user payment system” implemented by Japan [5] have all given increasingly more attention to the importance of reverse logistics and circular economy, and construction waste recycling in these countries is also effective. As the first country to propose recycling C&D waste, Germany has promulgated dozens of regulations on waste management since 1960. In the United States, the recycling of C&D waste also started early. The “Solid Waste Disposal Act” promulgated in 1965 involved many aspects of waste recycling, such as recycling regeneration and recycling standards. In Europe, due to the huge volume of C&D waste, the EU Directive “2008/98/EC” emphasizes that the EU urgently needs to improve the recycling rate of C&D waste [6]. In contrast, China’s C&D waste recycling activities started late. Statistical data show that the recycling rate of C&D waste is extremely low in China, only 5% [7]. There is still a large gap compared with other foreign countries.

The enormous volume of C&D waste, the high difficulty of recycling, and the lack of systematic coordination and management are the main reasons for the slow progress and low efficiency of C&D waste recycling in China. In recent years, China has promulgated a number of laws and regulations related to the recycling and remanufacturing of C&D waste. Construction materials supply-chain members are encouraged to participate in the recycling and remanufacturing activities [8]. Under this background, third party recycling agencies (recyclers) emerged as the times require, give full play to their professional advantages, improve the quality of the waste recycling, shorten the recycling time, and make the recycling activities more thorough and efficient. These measures have promoted the resource utilization of C&D waste in China to a certain extent, but the expected effect has not been achieved, and the resource utilization rate of C&D waste in China is still low. Through comprehensive analysis, we find that current construction contractors mainly recycle C&D waste by themselves [9]. Instead of solving problems from the overall perspective, they only refer to their own development indicators. Thus, their enthusiasm of recycling C&D waste is generally low and lacks motivation, which results in incomplete recycling of C&D waste [10]. The quality of recycled building materials cannot be guaranteed. For recyclers, the application scope of renewable building materials is narrow, and sales volume is relatively limited due to the low social acceptance of renewable building materials and the high cost of C&D waste recycling. As a result, the income of recyclers can barely cover the cost, and it is difficult for recyclers to compete in the market [11]. Meanwhile, in order to save cost and save trouble, many construction contractors continue to try every means to avoid supervision and choose illegal clearance or a low-price landfill [12]. It eventually led to a vicious cycle in C&D waste recycling.

Based on the analysis of all the difficulties existing in China’s C&D waste recycling, it is not difficult to conclude that managing the problems from the overall perspective plays an important role in C&D waste recycling. In order to promote the process of C&D waste recycling smoothly, the government and the enterprises should work together and make joint efforts. Facing the severe situation of C&D waste recycling, finding a solution is particularly important.

Our work is closely related to the stream of C&D waste recycling, the application of game theory in green supply-chain management, and the impact of governmental intervention on green supply-chain research. Currently, the importance of C&D waste management is increasingly adopted by academics. Increasingly more researchers have focused on studying C&D waste management. Nagapan et al. [13] explored the impacts of C&D waste on sustainable development and formulated several ways to reduce waste. Mohamed et al. [14] established three quantitative models to analyze the environmental and economic benefits of waste management, and found that recycling is the most environmentally friendly method of C&D waste management. Coronado et al. [15] developed a two-step methodology for the quantification and management analysis of C&D waste. Bilal et al. [16] investigated the potential of BIM (building information modeling) for C&D waste minimization and studied the usage of big data technologies in the context of C&D waste management. Chen et al. [17] and Kim et al. [18] have also studied the applicability of emerging digital technologies such as big data and BIM in C&D waste management. Liu et al. [19] concluded that the government should develop the recycling consciousness of the public and encourage technological innovation to improve environmental benefits of C&D waste management. These scholars have conducted extensive research on C&D waste recycling from the perspective of BIM. Although their contribution to C&D waste management is obvious, there are still some questions remaining unsolved. In the C&D waste recycling process, there are material flows and information flows among the participants. It can be regard as a supply chain and we can manage it using the concept of supply-chain operation management. Therefore, different from the aforementioned literature, we optimize the C&D waste management mode from the perspective of supply-chain operation management and study the selection of recycling channels of C&D waste.

In addition, there are papers that consider the application of game theory in green supply-chain management. The application of game theory in the management and decision-making of green supply chain is relatively mature. Scholars worldwide have used game theory to study the behavior evolution and decision-making of the participants under different power structures in green supply chain [20]. Chen et al. [21] studied a supply chain composed of remanufacturers, retailers, and consumers. By using game theory, they analyzed the pricing strategies under two recycling modes: direct recycling mode and indirect recycling mode. Cao et al. [22] constructed and solved an evolutionary game model of industrial solid waste recycling and remanufacturing between manufacturers and suppliers. The results showed the ratio of input and return on green recycling and remanufacturing affect recycling strategy. Shan et al. [23] employed the theory of the evolutionary game to analyze the problem in dynamic asymmetrical mutual cooperation, in which manufacturers and retailers share a part of the investment. Other than these two types of game theory, a Stackelberg game is also one of the most commonly used types. In a Stackelberg game, there are leaders and followers because the participants are not always in the same power status. The participants make decisions in a specific sequence; that is, the leaders make decisions first, and then the followers make their decisions based on the leaders’ decisions [24]. Zhang et al. [25] used a Stackelberg game to study the preference of supply-chain members for the decision-making power of energy efficiency level in the presence of cost-learning effect. Xu et al. [26] established a Stackelberg game model led by a manufacturer and studied the influence of green degree of products, wholesale prices, and retail prices on profits. Li et al. [27] studied a two-level green supply chain considering product green degree and established a Stackelberg game model to solve the equilibrium decision-making problems of supply-chain members. Shao et al. [28] considered a two-stage supply chain and developed Stackelberg game models to analyze the optimal decisions of supply-chain members and the contractual coordination in the supply chain considering government green subsidies. These scholars have helped decision-makers select the optimal choices and have also provided valuable management suggestions for supply-chain members.

Governmental regulation is an effective way to promote the healthy development of green supply chains. Governmental regulation is the basis for ensuring the members of green supply chains to perform their duties conscientiously. Mohammadreza et al. [29] studied the optimal decision of members of green supply chain with or without governmental intervention. The results indicated that the increase in the carbon tax imposed by the government significantly improves the green degree of products. Hafezalkotob et al. [30] used game theory to study the competition between a green supply chain and a regular supply chain under governmental economic tax. It was concluded that the green supply-chain members benefit from the increase in governmental tax on nonenvironmentally friendly products. Madani et al. [31] established a Stackelberg game model with the government as the leader and the green supply chain and the regular supply chain both as followers. The results showed that supply-chain members can produce more green products with governmental intervention. Chen et al. [32] used evolutionary game theory to study the behavior decisions of government and manufacturers under different carbon tax and subsidy mechanisms. Tian et al. [33] established a three-stage evolutionary game model of green supply-chain management under a governmental subsidy mechanism. It was found that the governmental subsidy mechanism can promote the increase in green product sales and encourage more enterprises to implement green supply-chain management. The research of these scholars shows that governmental intervention is very important for enterprises to implement environmentally friendly management, and the role of government in green supply-chain management should not be underestimated.

To summarize, through the review of the existing literature, it can be seen that few scholars have studied C&D waste recycling from the perspective of supply-chain management. Additionally, most existing research is focused on the single channel recycling of C&D waste, rarely involving the selection of recycling channels. This paper innovatively proposes the dual-channel recycling problem of C&D waste from the perspective of supply-chain operation, aiming to study the impact of the selection of recycling channels and the government’s economic intervention on pricing decisions. Based on this, this paper focuses on the following issues for in-depth research and discussion:In C&D waste recycling activities, do construction contractors prefer direct or indirect channels? Under what conditions will the construction contractor choose indirect channels for recycling?Under what circumstances will the recycler take the initiative to participate in recycling activities?How should the government formulate corresponding regulations to encourage recyclers to actively participate in the recycling activities of C&D waste?

In order to solve these problems and obtain conclusions of managerial value, we research the problem of dual-channel construction and demolition waste recycling. In this paper, we build a three-echelon supply chain, including a construction contractor, a professional recycling agency (recycler), and a building materials manufacturer, considering two modes: direct channel and indirect channel. We use game theory to analyze the optimal decision strategies of the members with and without governmental intervention, and the equilibrium strategies are obtained. We aim to find an ideal supply-chain management mode in C&D waste recycling.

## 2. Model Overview

Considering the sustainability of economy and environment, this paper aims to analyze and discuss the optimal pricing decision of dual-channel C&D waste recycling under three-echelon supply chain based on game theory.

### 2.1. Notation

The definitions of symbols used in this article are shown in Table 1.

### 2.2. Model Structure of Three-Echelon Dual-Channel Supply Chain

In this paper, we focus on the C&D waste recycling problem from the perspective of supply-chain operation management. We establish a three-echelon supply chain including a construction contractor (known as contractor), a professional recycler (recycler), and a building materials manufacturer (manufacturer). We use game theory as a mathematical tool to establish and solve the model.

The contractor and the recycler can both recycle C&D waste, but they have quite different recycling capacities. The contractor can only do some simple recycling jobs, such as sorting, crushing, or comminuting C&D waste. However, the recycler can recycle C&D waste more efficiently and thoroughly than the contractor. That is because the recycler has advanced recycling machinery and mature recycling technology. In conclusion, if C&D waste is recycled by the contractor, it can only be used to remanufacture second-class renewable building materials with slightly inferior quality. If it is recycled by the recycler, the final product can be first-class renewable building materials with higher quality.

There are two recycling modes for supply-chain members to recycle C&D waste: the direct channel (directly recycling C&D waste) and the indirect channel (entrusting the recycling job to the professional recycling agency—recycler). After the process of recycling, C&D waste turns into the recycled materials—an intermediate product that needs to be further remanufactured. The recycled materials can be directly used for remanufacturing and sold to the manufacturer for reproduction.

The graphical structure model of this dual-channel supply chain is shown in Figure 1.

In the direct channel model, the contractor directly recycles the waste generated on the construction site, and sells the recycled materials to the manufacturer for reproduction. After remanufacturing, the manufacturer puts the second-class renewable building materials into the market. The decision-making process is as follows: the contractor first determines the price of the recycled materials, and then the manufacturer determines the price of the second-class renewable building materials.

In the indirect channel model, the contractor sells C&D waste to the recycler at a certain price. After recycling the C&D waste, the recycler sells the recycled materials to the manufacturer for reproduction. Finally, the manufacturer puts the first-class renewable building materials into the market. The decision-making process of these three parties is as follows: Initially, the contractor determines the price of the C&D waste. Next, the recycler determines the price of recycled materials, and finally, the manufacturer determines the price of the first-class recycled building materials.

### 2.3. Assumptions

(1)To simplify the problem, we only consider the circumstance of one contractor, one recycler, and one manufacturer.(2)The contractor, the recycler, and the manufacturer are independent and rational decision-makers with the goal of maximizing their own interests.(3)The contractor and the recycler have different recycling capacities and apparently, the recycler does a better job. Thus, we assume that we can obtain second-class renewable building materials through the direct channel and we will obtain first-class renewable building materials through the indirect channel.(4)Consumers’ demand for final products is a linear function of the final retail price and the demand for the final product (including first-class and second-class renewable building materials) is always not negative.(5)Referring to the similar methods of Savaskan et al. [35], it is assumed that the fixed cost “*M*” of C&D waste recycling is directly proportional to the square of “*θ*”, i.e., *M**_Ι_* = *mθ**_Ι_*^2^, or *M**_Ι_**_Ι_* = *mθ**_ΙΙ_*^2^ (m is the difficulty coefficient of recycling C&D waste).

## 3. Game Model

In this section, several game models under different scenarios are established to solve the optimal decisions of supply-chain members. We choose to use Stackelberg game because the chain members are not in the same power status. Additionally, we use the reverse induction method to solve the game models.

### 3.1. Decision-Making Models without Governmental Intervention

#### 3.1.1. Case 1: Direct Channel (Contractor Singularly Recycles C&D Waste)

In this scenario, the contractor directly recycles the C&D waste that was generated on the construction site. We build a sequential noncooperative Stackelberg game considering the contractor as the leader and the manufacturer as the follower. Firstly, the contractor sets the price of the recycled materials to the manufacturer, and secondly, the manufacturer sets the price of the renewable building materials to the customers.

Their profit functions can be expressed as revenues minus costs:

Expected profit of the contractor = sales revenue of recycled materials − cost of recycling the C&D waste;

Expected profit of the manufacturer = sales revenue of renewable building materials − cost of purchasing of recycled materials − cost of remanufacturing.

In this case, the Stackelberg game model is as follows:(1)$$Level\;1:\;max\;\pi_{C}\left(P_{C}\right)=Q_{II}\left(P_{C}-C_{C}\right)-m\theta_{II}{}^{2}\;\;\;\;\;\;s.t.P_{C}\geq C_{C},\;Q_{II}\geq 0$$
(2)$$Level\;2:\;max\;\pi_{M}\left(P_{II}\right)=Q_{II}\left(P_{II}-P_{C}-C_{II}\right)\;s.t.\;P_{II}\geq P_{C}+C_{II},\;Q_{II}\geq 0$$

The reverse induction method is used to solve the game model. That is, when the price *P_C_* of C&D waste set by the contractor is determined, the manufacturer’s optimal decision *P**_ΙΙ_** can be obtained first. Since the object function is concave in nature, we can obtain the best response function for unit price by solving the first-order conditions. The optimal decision price *P**_ΙΙ_** is as follows:(3)$$P_{II}=\frac{\alpha +P_{C}+C_{II}}{2}$$

After that, we substitute the manufacturer’s best response function into the contractor’s profit function. It is also concave in nature and there is a maximum. The optimal value of contractor’s decision variable can be obtained by solving the first-order conditions:(4)$$P_{C}{}^{*}=\frac{\alpha {+C}_{C}-C_{II}}{2}$$

Substituting Formula (4) into Formula (3), we can obtain the manufacturer’s optimal decision price.

**Proposition** **1.**
*The optimal decision price of the contractor and manufacturer is as follows:*

$$\{\begin{array}{c} P_{C}{}^{*}=\frac{\alpha {+C}_{C}-C_{II}}{2}\\ P_{II}{}^{*}=\frac{3\alpha {+C}_{C}+C_{II}}{4} \end{array}$$



**Proposition** **2.**
*By substituting the manufacturer’s and the contractor’s optimal decision price into their profit function, respectively, the profits of the manufacturer and the contractor under the optimal decision-making situation are obtained:*

$$\{\begin{array}{c} \pi_{C}{}^{*}=\frac{{\left(\alpha -C_{C}-C_{II}\right)}^{2}}{8}-m\theta_{II}{}^{2}\\ \pi_{M}{}^{*}=\frac{{\left(\alpha -C_{C}-C_{II}\right)}^{2}}{16} \end{array}$$



**Proposition** **3.**
*According to Propositions 1 and 2, the profits of the contractor and the manufacturer are positively correlated with the basic market size of second-class renewable building materials α, and negatively correlated with the recycling cost of C&D waste C_C_ and the remanufacturing cost of second-class renewable building materials C*
*_ΙΙ_. Furthermore, the profit of contractor is also negatively related to the recycling difficulty coefficient m and recycling degree θ*
*_ΙΙ_.*


#### 3.1.2. Case 2: Indirect Channel (Contractor Entrusting the Recycling Job to the Recycler)

In this scenario, the contractor entrusts the C&D waste recycling job to the recycler. The contractor acts as the leader, and the recycler and the manufacturer act as the followers to conduct a sequential noncooperative Stackelberg game. Initially, the contractor sets the price of C&D waste for the recycler. Next, the recycler sets the price of the recycled materials to the manufacturer. Finally, the manufacturer sets the retail price of the renewable building materials.

Similarly, their profit functions can be obtained:

Expected profit of the contractor = sales revenue of C&D waste;

Expected profit of the recycler = sales revenue of recycled materials − cost of purchasing of C&D waste − cost of recycling C&D waste;

Expected profit of the manufacturer = sales revenue of renewable building materials − cost of purchasing of recycled materials − cost of remanufacturing.

In this case, the Stackelberg game model is as follows:{(5)Level 1: max πC(P0)=P0QIθI      s.t. P0≥0(6)Level 2: max πR(PR)=QI(PR−CR)−mθI2−P0QIθI      s.t. PR≥CR(7)Level 3: max πM(PI)=QI(PI−PR−CI)      s.t. PI≥PR+CIs.t. QI≥0 QI/θI≥0 

The model is also solved by the reverse induction method. Since the manufacturer’s profit is a concave function of retail price, the optimal retail price of the manufacturer can be obtained by solving the first-order conditions:(8)$$P_{I}=\frac{\beta +P_{R}+C_{I}}{2}$$

By substituting Formula (8) into the profit function of the recycler, the optimal decision price of the recycler can be obtained:(9)$$P_{R}=\frac{\theta_{I}\left(\beta +C_{R}-C_{I}\right)+P_{0}}{2\theta_{I}}$$

By substituting Formulas (8) and (9) into the contractor’s profit function, the contractor’s optimal decision price can be obtained:(10)$$P_{0}{}^{**}=\frac{\theta_{I}\left(\beta -C_{R}-C_{I}\right)}{2}$$

**Proposition** **4.**
*The optimal decision price of the contractor, the recycler, and the manufacturer can be obtained as follows:*

$$\{\begin{array}{c} P_{0}{}^{**}=\frac{\theta_{I}\left(\beta -C_{R}-C_{I}\right)}{2}\\ P_{R}{}^{**}=\frac{3\beta +C_{R}-3C_{I}}{4}\\ P_{I}{}^{**}=\frac{7\beta +C_{R}+C_{I}}{8} \end{array}$$



**Proposition** **5.**
*By substituting the manufacturer’s, the contractor’s, and the recycler’s optimal decision price into their profit function, respectively, the profits of the manufacturer, the contractor, and the recycler under the optimal decision-making situation are obtained:*

$$\{\begin{array}{c} \pi_{C}{}^{**}=\frac{{\left(\beta -C_{R}-C_{I}\right)}^{2}}{16}\\ \pi_{R}{}^{**}=\frac{{\left(\beta -C_{R}-C_{I}\right)}^{2}}{32}-m\theta_{I}{}^{2}\\ \pi_{M}{}^{**}=\frac{{\left(\beta -C_{R}-C_{I}\right)}^{2}}{64} \end{array}$$



**Proposition** **6.**
*According to Propositions 4 and 5, we can know that the profits of the contractor, the recycler, and the manufacturer are positively correlated with the basic market size of first-class renewable building materials β, and negatively correlated with the recycling cost of C&D waste C_R_ and remanufacturing cost of first-class renewable building materials C_I_. Meanwhile, the profit of the recycler is also negatively correlated with the recycling difficulty coefficient m and the recycling degree θ_I_.*


### 3.2. Decision-Making Models with Governmental Intervention

Because of increasingly prominent environmental problems, environmental protection has been given increasingly more attention. The recycling of C&D waste is one of the important measures of environmental protection. With improving the recycling amount of C&D waste as a basic goal, the government intends to implement an economic-reward mechanism, and apparently, the recycler always performs a better job than the contractor when recycling C&D waste. Thus, the target of the governmental subsidy is the recycler. In order to encourage the recycler to recycle and reuse C&D waste, the government subsidizes the recycler based on the recycling amount. The unit subsidy amount is *t*(*t* > 0); that is, the government gives subsidies *Qt* to the recycling amount *Q* of recycler.

Therefore, in the case of governmental intervention (case 3), in the indirect channel with the participation of recycler, the profits of supply-chain members are as follows:{(11)Level 1: max πC(P0)=P0QIθI      s.t. P0≥0(12)Level 2: max πR(PR)=QI(PR−CR)−mθI2−P0QIθI+tQI      s.t. PR≥CR(13)Level 3: max πM(PI)=QI(PI−PR−CI)      s.t. PI≥PR+CIs.t. QI≥0, QI/θI≥0 

We also use the reverse induction method to solve the model. At first, we obtain the best response function of manufacturer:(14)$$P_{I}=\frac{\beta +P_{R}+C_{I}}{2}$$

By substituting Formula (14) into the profit function of the recycler, the best response function of the recycler can be obtained:(15)$$P_{R}=\frac{\theta_{I}\left(\beta +C_{R}-C_{I}-t\right)+P_{0}}{2\theta_{I}}$$

By substituting Formulas (14) and (15) into the contractor’s profit function, the contractor’s optimal decision price can be solved:(16)$$P_{0}{}^{***}=\frac{\theta_{I}\left(\beta -C_{R}-C_{I}+t\right)}{2}$$

**Proposition** **7.**
*By solving these equations simultaneously, the optimal decision price of the contractor, recycler, and manufacturer can be obtained:*

$$\{\begin{array}{c} P_{0}{}^{***}=\frac{\theta_{I}\left(\beta -C_{R}-C_{I}+t\right)}{2}\\ P_{R}{}^{***}=\frac{3\beta +C_{R}-3C_{I}-t}{4}\\ P_{I}{}^{***}=\frac{7\beta +C_{R}+C_{I}-t}{8} \end{array}$$



**Proposition** **8.**
*Substituting the optimal decision prices of the three into their respective profit functions, the profit of the contractor, recycler, and manufacturer under the optimal decision situation can be obtained:*

$$\{\begin{array}{c} \pi_{C}{}^{***}=\frac{{\left(\beta -C_{R}-C_{I}+t\right)}^{2}}{16}\\ \pi_{R}{}^{***}=\frac{{\left(\beta -C_{R}-C_{I}+t\right)}^{2}}{32}-m\theta_{I}{}^{2}\\ \pi_{M}{}^{***}=\frac{{\left(\beta -C_{R}-C_{I}+t\right)}^{2}}{64} \end{array}$$



**Proposition** **9.**
*From Proposition 7 and 8, it can be seen that the profits of the contractor, recycler, and manufacturer are positively correlated with the basic market size of the first-class renewable building materials β and the amount of governmental subsidies t, and negatively correlated with the recycling cost of C&D waste C_R_ and the remanufacturing cost of first-class renewable building materials C_I_, Furthermore, the profit of the recycler is also negatively correlated with recycling difficulty coefficient m and recycling degree θ_I_.*


## 4. Model Analysis

### 4.1. Comparison of Scenario 1 and Scenario 2

By comparing the profits of the manufacturer, contractor, and recycler under the two situations in which the contractor recycles C&D waste alone and the contractor entrusts the recycling job to the recycler (Proposition 2 with Proposition 5), Table 2 below can be obtained:

It can be seen from Table 2 that if the profit of the supply-chain members in scenario 2 is not lower than that in scenario 1, then supply-chain members are more likely to choose scenario 2, that is, to recycle C&D waste through the indirect channel.

For the contractor, if *π_C_*** ≥ *πC**, the contractor is more willing to entrust the recycler to recycle C&D waste rather than recycling C&D waste alone. We can show that *π_C_*** − *π_C_** ≥ 0, i.e., (*β* − *C_R_* − *C_I_*)^2^/16 − [(*α* − *C_C_* − *C_II_*)^2^/8 − *mθ_II_*^2^] ≥ 0 is satisfied. For the recycler, if and only if *π_R_*** ≥ 0, the recycler is willing to participate in the C&D waste recycling activities. At this time, *(β* − *C_R_* − *C_I_)^2^*/32 − *mθ_I_*^2^ ≥ 0 is satisfied. For the manufacturer, if *π_M_*** ≥ *π_M_**, the manufacturer is more willing to purchase the recycled materials from the recycler. At this time, *π_M_*** − *π_M_** ≥ 0 is satisfied, that is (*β* − *C_R_* − *C_I_*)^2^/64 − (*α* − *C_C_* − *C_II_*)^2^/16 ≥ 0 is satisfied.

When (*α* − *C_C_* − *C_II_*)^2^/8 − *mθ_II_*^2^ < 0 is satisfied, the contractor’s profit in scenario 1 is negative, that is, the contractor will not choose the direct channel. At this time, if (*β* − *C_R_* − *C_I_*)^2^/32 − *mθ_I_*^2^ ≥ 0, then the contractor, the recycler, and the manufacturer will choose the indirect channel to recycle C&D waste.

**Theorem** **1.***When (α* − *C_C_* − *C_II_)^2^/8* − *mθ_II_^2^* < *0 and (β* − *C_R_* − *C_I_)^2^*/*32* − *mθ_I_^2^* ≥ *0 are both satisfied, supply-chain members will choose scenario two (the indirect channel) to carry out C&D waste recycling activities.*

When (*α* − *C_C_* − *C_II_*)^2^/8 − *mθ_II_*^2^ > 0 is satisfied, the profits of the contractor and the manufacturer in both scenario 1 and 2 are non-negative. Only when the profit in scenario 2 is higher than that in scenario 1, the contractor and manufacturer will choose the indirect channel in scenario 2. At this time, only when the profit of the recycler is also not negative can we make certain that the indirect channel can be implemented smoothly.

**Theorem** **2.**
*Only when the following conditions are met will the supply-chain members choose the indirect channel in scenario 2 to recycle C&D waste:*

$$\{\begin{array}{c} \frac{{\left(\alpha -C_{C}-C_{II}\right)}^{2}}{8}-m\theta_{II}{}^{2}\geq 0\\ \frac{{\left(\beta -C_{R}-C_{I}\right)}^{2}}{16}\geq \frac{{\left(\alpha -C_{C}-C_{II}\right)}^{2}}{8}-m\theta_{II}{}^{2}\\ \frac{{\left(\beta -C_{R}-C_{I}\right)}^{2}}{32}-m\theta_{I}{}^{2}\geq 0\\ \frac{{\left(\beta -C_{R}-C_{I}\right)}^{2}}{64}\geq \frac{{\left(\alpha -C_{C}-C_{II}\right)}^{2}}{16} \end{array}$$



According to theorem 1 and 2, when some certain parameters conditions are met, each member of the supply chain will choose the indirect channel to recycle C&D waste. If supply-chain members choose the indirect channel to recycle C&D waste, their profits should be no less than those in the direct channel. When the basic market scale of the first-class renewable building materials increases, the members of supply chain are more likely to choose the indirect channel. Furthermore, decreasing the unit recycling cost and remanufacturing cost can increase the profits of supply-chain members and the total profit of the system.

### 4.2. Comparison of Scenario 2 and Scenario 3

When the contractor entrusts the recycling job to the recycler, by comparing the profits and the optimal decision price of the manufacturer, the contractor and the recycler under the two situations in which decision-making models without governmental intervention and decision-making models with governmental intervention (Propositions 4 and 5 with Propositions 7 and 8), Table 3 below can be obtained:

Comparing the retail price of first-class renewable building materials in scenario 2 and scenario 3, it is easy to show that the retail price of first-class renewable building materials in scenario 3 is lower than that in scenario 2.

**Theorem** **3.**
*With governmental intervention, the retail price of first-class renewable building materials will decrease and customers can benefit from it.*


By comparing the profits of every supply-chain member and the total profit of the whole system in scenario two and three (as shown in Table 3), the profits in scenario three are always higher than those in scenario two.

**Theorem** **4.**
*In the scenario with governmental intervention, the profits of each member in the supply chain and the total profit of the system increase rapidly.*


By comparing the recycling amount of C&D waste in scenario two and three, we can easily show that the recycling amount of C&D waste in scenario 3 is much higher than that in scenario 2.

**Theorem** **5.**
*In the scenario with governmental intervention, the recycling amount of C&D waste increases rapidly.*


From Theorems 3–5, we can conclude that governmental intervention is very feasible and effective. The governmental incentive subsidy increases the profits of all the supply-chain members, and also stimulates the enthusiasm of enterprises to participate in the recycling activities of C&D waste. It also reduces the retail price of renewable building materials, which benefits the majority of consumers. Additionally, the recycling amount of C&D waste increases when the governmental intervention exists, which contributes to environmental protection!

## 5. Sensitivity Analysis

In order to analyze and verify the model, a numerical study is carried out to examine the impact of various parameters on the model output in this section. By using Mathematica software for simulation, we verify the accuracy of the above conclusions and explore the impact of the governmental regulations and other important parameters on decision variables and members’ profits. The parameter setting is based on the field investigation of two typical building material manufacturers in Chongqing: where α = 1100; β = 2200; C_II_ = 20; C_I_ = 30; C_C_ = 10; C_R_ = 25; *θ*_II_ = 0.3; *θ*_I_ = 0.8; M = 2000; t = 100.

It can be seen from Table 4 that under the given parameters, the profit of the contractor in scenario 2 is much greater than that in scenario 1. In this case, to earn more and obtain a higher profit, the contractor entrusts the recycler to recycle C&D waste. Furthermore, the profits of the three parties reach the highest in scenario 3, which indicates that the governmental incentive policy is feasible and effective. The conclusions here are consistent with the theoretical analysis.

To further analyze the impact of the governmental regulations and other important factors on the decision-making strategy of supply-chain members, we use Wolfram’s Mathematica software for sensitivity analysis.

It can be seen from Figure 2 and Figure 3 that the profits of both the contractor and manufacturer decrease with the increase in remanufacturing cost and recycling cost. The decrease trend of the contractor’s profit is more obvious with the recycling cost. Thus, it can be seen that if the difficulty of recycling and cost increases, the contractor’s enthusiasm for recycling is reduced and the recycling amount of C&D waste decreases rapidly. As a result, there are insufficient recycled materials for the manufacturer to remanufacture. Therefore, it is not enough if only the contractor participates in C&D waste recycling activities.

Figure 4 shows that when the recycling difficulty coefficient and the recycling degree of C&D waste increase, the contractor’s profit decreases, and the profit is more sensitive to the recycling degree. With the increase in the recycling degree, the profit drops sharply. That is because that the contractor lacks professionalism and technicality in C&D waste recycling work. The labor and material cost required to improve unit recycling degree is relatively high. Therefore, if the contractor recycles C&D waste alone, it often leads to an incomplete and low recycling level. As a result, the quality of the renewable building materials cannot be guaranteed.

From Figure 5 and Figure 6, we can know that the price and profit increase with the increase in the market scale of renewable building materials. The increase in the market scale of renewable building materials means the increase in the consumers’ WTP (willingness to pay) of renewable products. The growing public awareness of environmental protection will boost demand for the renewable products, and the purchase of renewable building materials will increase. However, these hardly change with the recycling cost and the curve trend is mostly stable. It can be seen that for recyclers, due to advanced technology and professionalism, the change in recycling cost has little impact on price and profit. The increase in market scale of renewable building materials will attract and encourage more enterprises to participate in C&D waste recycling activities.

Figure 7 and Figure 8 both reflect that with the increase in governmental subsidies, the retail price of renewable building materials will decrease smoothly, which means consumers can pay less, and the demand for renewable building materials will increase. The rising demand for renewable building materials will accelerate the process of C&D waste recycling, and as a result, the recycling amount of C&D waste will be increased. At the same time, with the increase in subsidies, the profits of the three parties will increase. It can be seen that governmental subsidies can simultaneously contribute to environmental protection and increased social welfare. In addition, from Figure 8, it can be seen that the profit of the contractor is always the largest and its sensitivity to government subsidies is the highest, which indicates that the contractor is more willing to entrust recycling to the recycler when governmental subsidies are increasing, which once again proves that the governmental incentive policy is very feasible and effective.

## 6. Conclusions

In this paper, in order to improve the recycling efficiency of C&D waste and promote the process of C&D waste management, we study the decision-making problem of a dual-channel, three-echelon supply chain of C&D waste. We use game theory to analyze the optimal decision strategies of members with and without governmental intervention, and the equilibrium strategies are obtained. The members’ behavior strategies in these three models are discussed, based on the example of Chongqing; the model of this study was applied to carry out numerical simulations of the optimal decision strategy for the government and enterprises in construction waste recycling process. The results reveal that a larger recycling cost of the C&D waste leads to lower profits for the members. When the market capacity of first-class renewable building materials increases, the recycler’s recycling cost of C&D waste decreases, and the remanufacturing cost of recycled materials processed into first-class renewable building materials decreases, the supply-chain members will choose the indirect channel to do the recycling. In the case when government intervention exists, the recycling quantity of C&D waste increases, the price of the final products decreases, and both the total profit of the system and the profit of the supply-chain members increase, and there is a positive correlation with the government subsidies. We offer references for effectively promoting recycling technological development in underdeveloped areas, and promoting C&D waste recycling and mitigating environmental degradation. Here, we obtain the following conclusions, which can provide managerial advice for government and supply-chain members.

(1) The contractor, the recycler, and the manufacturer should take the social responsibility and have a positive attitude toward C&D waste recycling. In order to reduce the unit recycling cost and the recycling difficulty coefficient of C&D waste, the recycler should improve recycling machinery and equipment, and learn and master advanced recycling technology. Only in this way can we eventually improve the recycling efficiency and increase the recycling amount of C&D waste. Furthermore, the manufacturer should actively promote the application of scientific and technological remanufacturing achievements to reduce the unit remanufacturing cost and increase the output of renewable building materials. Through the joint efforts of both the upstream and downstream of the supply chain, the profit of each member will be continuously improved. According to Figure 2 and Figure 3, by reducing costs, the profits of both contractors and remanufacturers increased to varying degrees.

(2) The government should let consumers know more about environmentally friendly products through public service advertisements and enhance the social recognition of renewable products. The public awareness of environmental protection will boost demand for the renewable products. The consumers’ WTP (willingness to pay) for renewable products will be improved and the purchase of renewable building materials will be increased. Furthermore, increasing the basic scale of the renewable product market is also a good idea because it can improve the profits of the supply-chain members. It will make the enterprises sense that it is profitable and engaging to join the C&D waste recycling activities. To form a virtuous circle, let more enterprises and the public change their negative attitude and instead participate in C&D waste recycling activities, which is beneficial to environmental protection and social welfare.

(3) The government should implement incentive policies for the recycler to create a good market competition environment and minimize the risk of the recycler withdrawing from the market. The governmental incentive policies can not only increase the profits of supply-chain members, but also stimulate the enthusiasm of the enterprises to participate in the C&D waste recycling activities. Additionally, there is a positive correlation between the recycling amount of C&D waste and the amount of governmental subsidy. With the increase in governmental subsidy, the recycling amount of C&D waste increases rapidly. Furthermore, governmental intervention can reduce the retail price of the first-class renewable building materials, which can benefit the majority of consumers, and achieve a win–win situation of economy, environment, and society.

Although this paper has drawn some insights on managerial significance to C&D waste recycling management, there are still some limitations. First, it is assumed that information is available to all members, which can be extended to asymmetric information settings. Additionally, it is also very meaningful to study a supply chain that is led by the contractor or the recycler. Moreover, in this study, the governmental regulation is static. However, a dynamic regulation environment is closer to reality and worth studying. Finally, this paper only considers one contractor, one recycler, and one manufacturer. Future research may consider multiple participants and establish a supply-chain network.

## Figures and Tables

**Figure 1 ijerph-19-04975-f001:**
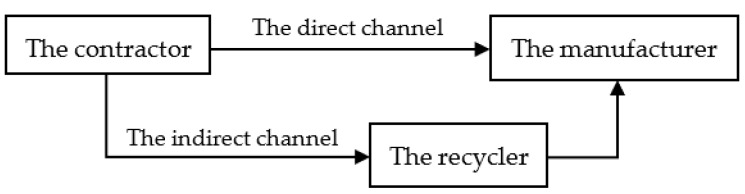
The graphical structure of the dual-channel supply chain.

**Figure 2 ijerph-19-04975-f002:**
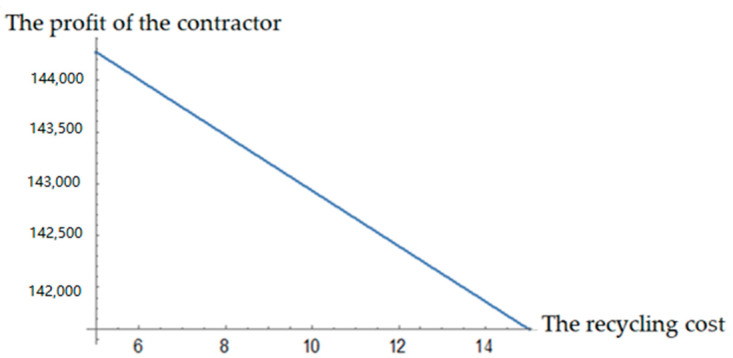
Effects of recycling cost on the contractor’s profit.

**Figure 3 ijerph-19-04975-f003:**
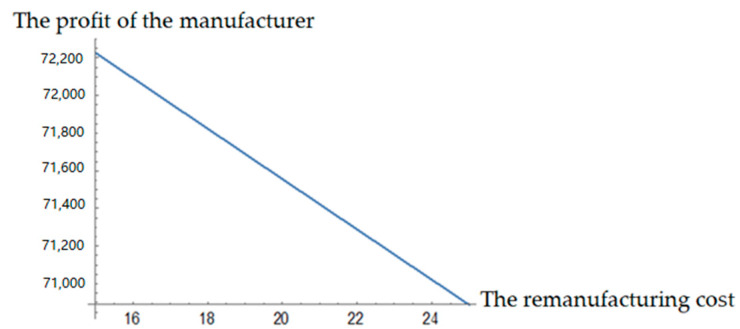
Effects of remanufacturing cost on the manufacturer’s profit.

**Figure 4 ijerph-19-04975-f004:**
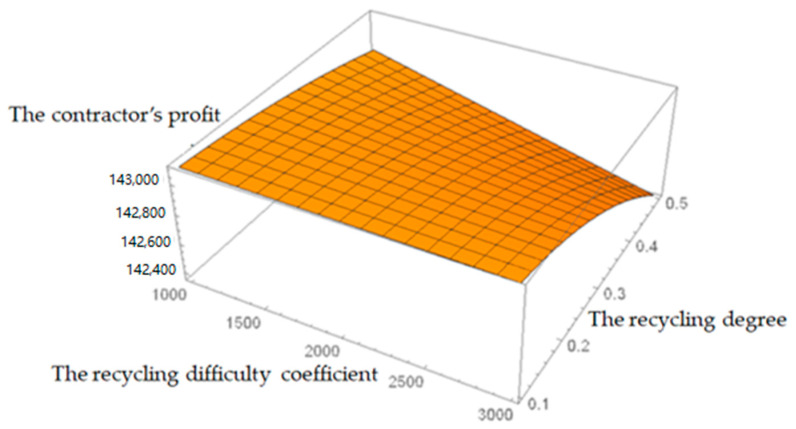
Effects of recycling difficulty coefficient and the recycling degree on the contractor’s profit.

**Figure 5 ijerph-19-04975-f005:**
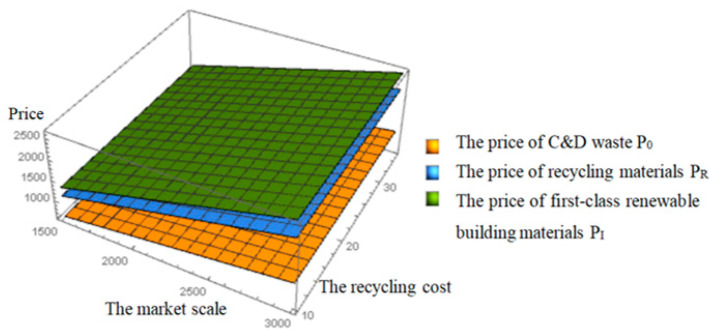
Effects of the market scale and the recycling cost on the prices.

**Figure 6 ijerph-19-04975-f006:**
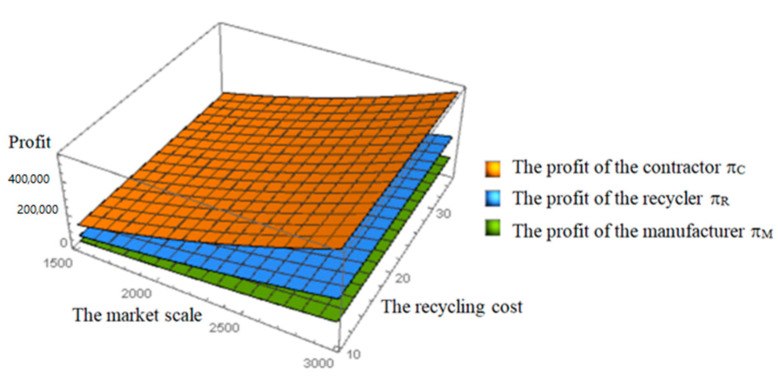
Effects of the market scale and the recycling cost on the profits.

**Figure 7 ijerph-19-04975-f007:**
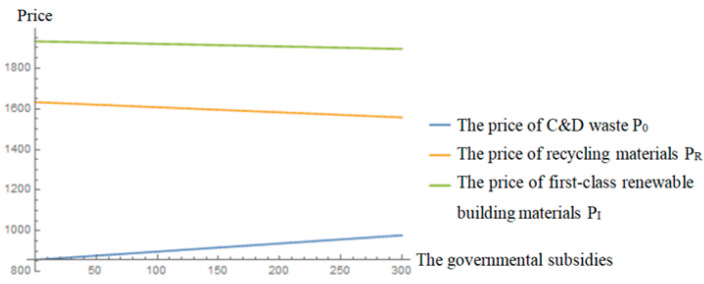
Effects of governmental subsidies on the prices.

**Figure 8 ijerph-19-04975-f008:**
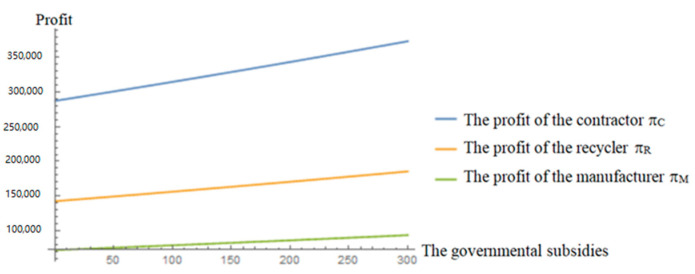
Effects of governmental subsidies on the profits.

**Table 1 ijerph-19-04975-t001:** Notation and definitions.

Notation	Definition	Unit
α	The market capacity of second-class renewable building materials, with α>0	/
β	The market capacity of first-class renewable building materials, with β>0	/
$P_{0}$	The intermediate transfer price of C&D waste made by the contractor in indirect channel, which is also the contractor’s decision variable	CNY/ton
$P_{I}$	The price of the first-class renewable building materials, the decision variable of the manufacturer in the indirect channel	CNY/ton
$P_{II}$	The price of the second-class renewable building materials, the decision variable of the manufacturer in the direct channel	CNY/ton
$P_{C}$	The price set by the contractor for the recycled materials in the direct channel, which is the contractor’s decision variable	CNY/ton
$P_{R}$	The price set by the recycler for the recycled materials in the indirect channel, which is the decision variable of the recycler	CNY/ton
$C_{I}$	The remanufacturing cost of recycled materials processed into first-class renewable building materials by manufacturer in indirect channel	CNY/ton
$C_{II}$	The remanufacturing cost of recycled materials processed into second-class renewable building materials by manufacturer in direct channel	CNY/ton
$C_{C}$	The contractor’s recycling cost of C&D waste in direct channel	CNY/ton
$C_{R}$	The recycler’s recycling cost of C&D waste in indirect channel	CNY/ton
$Q_{I}$	Market demand of first-class renewable building materials, similarly, with $Q_{I}=\beta -P_{I}$	ton
$Q_{II}$	Market demand of second-class renewable building materials, according to Ferrer et al. [34], with $Q_{II}=\alpha -Q_{II}$	ton
$\theta_{I}$	The recycling degree of C&D waste by the recycler in indirect channel or the efficiency of C&D waste recycling by the recycler, with $\;0<\theta_{I}<1$	/
$\theta_{II}$	The recycling degree of C&D waste by the contractor in direct channel or the efficiency of C&D waste recycling by the contractor, with $\;0<\theta_{II}<\theta_{I}<1$	/
m	The difficulty coefficient of C&D waste recycling, an exogenous variable which is related to the type of C&D waste, with 0<m<1, the larger m is, the more cost of recycling the C&D waste	/
$\pi_{C}$	Profit of the contractor	CNY/ton
$\pi_{R}$	Profit of the recycler	CNY/ton
$\pi_{M}$	Profit of the manufacturer	CNY/ton

**Table 2 ijerph-19-04975-t002:** Comparison of profits in scenario 1 and scenario 2.

		Contractor’s Profit	Recycler’s Profit		Manufacturer’s Profit

Scenario 1		*π*_C_* = (*α* − *C_C_* − *C_II_*)^2^/8 − *mθ_II_*^2^			*π*_M_* = (*α* − *C_C_* − *C_II_*)^2^/16

Scenario 2		*π*_C_** = (*β* − *C_R_* − *C_I_*)^2^/16	*π*_R_** = (*β* − *C_R_* − *C_I_*)^2^/32 − *mθ_I_*^2^		*π*_M_** = (*β* − *C_R_* − *C_I_*)^2^/64

**Table 3 ijerph-19-04975-t003:** Comparison of scenario 2 and scenario 3.

		Scenario 2	Comparison	Scenario 3

Price of building materials		*P_I_*** = (7*β* + *C*_R_ + *C_I_*)/8	≥	*P_I_**** = (7*β* + *C_R_* + *C_I_* − *t*)/8
Contractor’s profit		*π*_C_** = (*β* − *C*_R_ − *C_I_*)^2^/16	≤	*π_C_**** *=* (*β* − *C_R_* − *C_I_ + t*)^2^/16
Recycler’s profit		*π*_R_** = (*β* − *C*_R_ − *C_I_*)^2^/32 − *mθ_I_*^2^	≤	*π_R_**** *=* (*β* − *C_R_* − *C_I_ + t*)^2^/32 − *mθ_I_*^2^
Manufacturer’s profit		*π*_M_** = (*β* − *C*_R_ − *C_I_*)^2^/64	≤	*π*_M_*** = (*β* − *C_R_* − *C_I_ + t*)^2^/64
Recycling amount		*Q*** = (*β* − *C*_R_ − *C_I_*)/8*θ_I_*	≤	*Q**** = (*β* − *C_R_* − *C_I_ + t*)/8*θ_I_*
Total profit of the system		*π*** = 7(*β* − *C*_R_ − *C_I_*)^2^/64 − *mθ_I_*^2^	≤	*π**** = 7(*β* − *C_R_* − *C_I_ + t*)^2^/64 − *mθ_I_*^2^

**Table 4 ijerph-19-04975-t004:** Optimal profits under different scenarios.

Profit	Scenario 1		Scenario 2	Scenario 3
contractor	142,933		287,564	315,001
recycler			142,502	156,221

manufacturer	71,556		71,891	78,750

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
