# Peer review of "Analysis on Recycling Channel Selection of Construction and Demolition Waste in China from the Perspective of Supply Chain"

_ijerph, 2022, doi:10.3390/ijerph19094975_

Round 1
Reviewer 1 Report
This manuscript presents a theoretical analysis on channel selection of construction and demolition waste recycling from the perspective of supply chain. Advances in the domain have been achieved and the content of the case report is related to the scope of the journal but the author should consider the below comments to improve the paper.
- The author should reformulate the abstract in order to emphasize the novelty of the paper. More numerical results should be inserted to give a better insight on the results.
- Please add a list of nomenclature and the unit of measurements for the listed parameters in accordance with SI.
- Please insert a list of abbreviations which would be useful considering the notations used in the paper. It is enough to define abbreviations only once in the text do not define them repeatedly.
- Section 2 needs to be integrated into the introduction section. It is not a review article...
- Equations need to numbered and cited in the text.
- In my opinion the study is too theoretical and needs to be validated against experimental data. The authors need to provide experimental data, based on own findings or from the literature, to demonstrate the reliability and realistic approach of the model.
- Conclusions need to be revised and numerical data has to be incorporated in the conclusion section.
Author Response
Thank you for your comments on our manuscript. Those comments are all valuable and very helpful for revising and improving our paper, as well as the important guiding significance to our researches. We have studied comments carefully and have made correction which we hope meet with approval.
Last but not least, thanks for your professional advice and suggestions again, if you have any other questions regarding our revised manuscript, please feel free to contact us.
Best regards.

Reviewer 2 Report
The authors discussed an analysis on channel selection of C&D waste recycling using game theory applied on dual-channel three echelon supply chain model.
The research clearly shown a contribution in a pool of research of this area. However, there are some points need to be addressed.
1) the staement regarding three main issues discussed in this study need to be rewriten in a more concise and clear way. Especially the first sentence (linke 72-76). It is rather vague in the current presentation.
2) It is necessary to add some review regarding relationship between C&D waste management and GSCM. Thus the relevancy between section 2.1 with 2.2 and 2.3 will be clear.
3) Regarding the model development, notation in line 225, 226, 227 need to be described in more detail. It is not clear what is the recycling degree of contractor and recyclear mean?. how does it applied in practical use. Moreover the difficulty coefficient is not been described clearly. How does people deterimen the difficulty coefficient?
4) The motivation of using stackelberg game model due to the same power status between player is not well explained. Why authors consider all player has the same power?.
5) All the formulation not only line 301,301, 308, 311, etc. need to be numbered properly. In addition, they also need to be explained in certain level of details. In the current form, it is not clear which formula is being refered, whenever the author refer to a formula with number.
- for example line 312, subtitute formula (2) and formula (1) which one is it?
6) Please provide a descriptive information regarding each scenario, not only the tabel with comparison of formula that different between scenarios.
Author Response

(The authors gave the same response as above.)

Reviewer 3 Report
Dear Authors,
The paper Analysis on Channel Selection of Construction and Demolition Waste Recycling from the Perspective of Supply Chain by Ruwen Tan, Xiang Qing, Jingsong Yang, Jing Zhang and Deng Li is well suited for International Journal of Environmental Research and Public Health. The authors of this article analyzed the results of the present studies on waste recycling in construction and demolition process.
The paper is substantially good and maybe scientifically valuable. In reviewer opinion the practical possibilities of use are limited.
The paper contains parts in good order. The Abstract is really the summary of the article. The length of the abstract is good. Keywords – only lowercase letters should be used “construction and demolition waste”.
Introduction and literature review - explains the general background of the issue, possibly too little described. Many references to the articles have been mentioned, but an outline of their contents is basic.
The article presents scenarios and makes many assumptions. The presented analysis shows the possibilities of the analysis. However, for real calculations, extensive research should be carried out to establish the exact relationships and their numerical representation. This article should be considered as a starting point for detailed analyzes.
The bibliography does not contain any publication from the Journal of Environmental Research and Public Health. Are you sure there weren't any articles the authors could refer to?
The bibliography was probably created with the use of a citation manager. There were shortcomings. In many entries "[J]" or “C” appears after the title, this is unnecessary and should be deleted. Line 656 - the end of the text should be corrected.
The article was written enough well in English, is understandable for a reviewer, a person who does not speak English as a mother tongue.
Minor omissions in the text. Reviewer found: surname in uppercase (line 280), unnecessary space (line 13), so the text should be carefully checked.
All text in line 606 - in the opinion of the reviewer, it should not be included in the published article, it is obvious that it should be removed before publication.
Author Response

(The authors gave the same response as above.)

Reviewer 4 Report
- The approach of the article is not appropriate, from my point of view. The management of Construction and Demolition Waste is very complex and I do not consider that it can be analyzed with a mathematical model, such as game theory.
- In other Countries and Continents, there are Directives and Laws on Waste Management that should be consulted as part of the Overview and knowledge base.
- The study approach, analyzing the profit or economic benefits of the processes is unethical if what interests us is the preservation of the environment. It is a matter of social conscience and commitment, which must be encouraged in any case with bonuses for waste recycling. In this sense, the study must contemplate the importance of the Circular Economy in the Waste Management process, of which there are also laws and guidelines in this regard in other countries, which should be referenced and analyzed.
- The study has many limitations, as indicated by its authors in the conclusions. Construction and Demolition waste is very varied and complex, including dangerous waste, and should be considered. In addition, there are many factors and agents in the process, depending on the different types of materials, their applications and environmental impacts. The waste management of plastics is not the same as that of aggregates, for example.
- The conclusions are very generalist, having been the result of mathematical analysis, but not adapted to reality.
Thank you
Author Response

(The authors gave the same response as above.)

Round 2
Reviewer 1 Report
The authors addressed the mot of my comments. I have only the following observation. Please reduce the length of the introduction and conclusion sections. Also, please add more numerical data to the abstract and conclusion sections and backup the drawn conclusion with the obtained results.
Author Response
Thanks for your kind suggestion. We have reduced the length of the introduction and conclusion sections (page 2-3, line 93-148 of the revision; page 17, line 592-600 of the revision). At the same time, we have also revised the abstract and conclusion sections according to your suggestion (page 1, line 20-22 of the revision; page 15-16, line 543-570 of the revision).
Last but not least, thanks for your professional advice and suggestions again, if you have any other questions regarding our revised manuscript, please feel free to contact us.
Best regards.

Reviewer 2 Report
I have seen the manuscript and I saw the authors have addressed all my concerns. I would recommend acceptance of this manuscript.

Author Response
Thanks for your professional advice and suggestions again, if you have any other questions regarding our revised manuscript, please feel free to contact us.
Best regards.

Reviewer 4 Report
I thank the authors for the effort in answering and solving to all the considerations indicated in the review.
The manuscript has improved in its structure and content and the approach is clearer, as well as the conclusions obtained.
However, since the study focuses on the state of the recycling process in China, which is far behind the advances in other countries, as indicated by the authors (Firstly, C&D waste recycling is in the early stage of development and recyclers is hard to sustain in China…), I recommend that this aspect is indicated in the title. I mean that China should appear as the country of application of the study.
Thank you
Author Response
Thanks for your professional advice and recognition of our efforts to revise the article. We have considered your doubts again and modified the title of the article to better fit the actual situation of construction waste treatment in China (page 1, line 1-3 of the revision). The current situation of construction waste recycling process in China is relatively backward compared with some countries, which is also the purpose of our study, hoping to provide theoretical reference for construction waste management in China through our study.
Thanks again for the hard work of the editors and reviewers, if there are other shortcomings in the article, we will actively modify it.
Best regards.
